# ZNF750: A Novel Prognostic Biomarker in Metastatic Prostate Cancer

**DOI:** 10.3390/ijms24076519

**Published:** 2023-03-30

**Authors:** Manuela Montanaro, Massimiliano Agostini, Lucia Anemona, Elena Bonanno, Francesca Servadei, Enrico Finazzi Agrò, Anastasios D. Asimakopoulos, Carlo Ganini, Chiara Cipriani, Marta Signoretti, Pierluigi Bove, Francesco Rugolo, Benedetta Imperiali, Gerry Melino, Alessandro Mauriello, Manuel Scimeca

**Affiliations:** 1Department of Experimental Medicine, Tor Vergata Oncoscience Research (TOR), University of Rome Tor Vergata, 00133 Rome, Italy; 2Department of Surgical Sciences, Division of Urology, University of Rome Tor Vergata, 00133 Rome, Italy; 3San Carlo di Nancy Hospital, 00165 Rome, Italy; 4San Raffaele University, Via di Val Cannuta 247, 00166 Rome, Italy; 5Faculty of Medicine, Saint Camillus International University of Health Sciences, Via di Sant’Alessandro 8, 00131 Rome, Italy

**Keywords:** ZNF750, prostate cancer, metastasis, prognostic biomarker

## Abstract

Prostate cancer is the most frequently diagnosed cancer and the fifth leading cause of cancer death among men in 2020. The clinical decision making for prostate cancer patients is based on the stratification of the patients according to both clinical and pathological parameters such as Gleason score and prostate-specific antigen levels. However, these tools still do not adequately predict patient outcome. The aim of this study was to investigate whether ZNF750 could have a role in better stratifying patients, identifying those with a higher risk of metastasis and with the poorest prognosis. The data reported here revealed that ZNF750 protein levels are reduced in human prostate cancer samples, and this reduction is even higher in metastatic samples. Interestingly, nuclear positivity is significantly reduced in patients with metastatic prostate cancer, regardless of both Gleason score and grade group. More importantly, the bioinformatics analysis indicates that ZNF750 expression is positively correlated with better prognosis. Overall, our findings suggest that nuclear expression of ZNF750 may be a reliable prognostic biomarker for metastatic prostate cancer, which lays the foundation for the development of new biological therapies.

## 1. Introduction

The prostate is the male gland most affected by tumors, and prostate cancer is the most frequently diagnosed cancer and the fifth leading cause of cancer death among men in 2020 [1]. Remarkably, the lack of specific and sensitive markers often leads to overtreatment of prostate cancer which eventually develops into castration-resistant prostate cancer (CRPC). Several mechanisms for the development of CRPC have been proposed, including increased androgen receptor (AR) expression, AR mutation, emergence of AR splice variants, increased intra-tumoral steroid hormone synthesis, and modulation of co-factor activity [2]. At the first histological diagnosis, about 80% of prostate cancers are localized whereas 20% have spread from the primary mass to regional lymph nodes or to distant organs [1]. The presence of metastasis significantly influences the patients’ prognosis. In fact, the 5-year survival rate is about 100% for prostate cancer patients with localized mass and only 30% for those with evidence of tumor metastasis. Thus, the prevention of metastatic lesion formation, as well as the identification of new reliable biomarkers of the metastatic process, are considered a major health challenge. Given this, recent studies have proposed new markers and/or molecular targets for metastatic prostate cancers [3,4,5,6,7,8,9]. However, at present, early biomarkers of metastasis are not available for prostate cancer [10].

Zinc finger (ZNF) proteins are one of the most numerous groups of proteins in the whole human genome [11]. The general zinc finger structural organization is preserved by the zinc ion which arranges cysteine and histidine in different combinations [12]. Among the eukaryotic transcriptional factors, the zinc finger domain is one of the most common DNA-binding motifs found [12]. Originally, ZNFs were only identified for their DNA-binding domains; nevertheless, the subsequently discovered multiple and unique ZNF motifs allow ZNF proteins to bind a wide range of target molecules, including RNA, methylated DNA, and lipids [13]. This suggests their potential role in both physiological and pathological processes [12,14].

Among ZNF proteins, the human zinc finger (C2H2-type) protein ZNF750 is a transcription factor composed of an atypical C2H2 zinc finger motif in the amino terminal domain and two highly conserved PLNLS sequences that are involved in DNA binding and protein–protein interactions (Figure 1).

ZNF750, by inducing terminal keratinocyte differentiation genes and repressing epidermal progenitor genes, plays a key role in regulating epithelial homeostasis [15,16]. Indeed, mutations within the C2H2 zinc finger motif destroy the ability of ZNF750 to activate differentiation genes, and they have been reported in patients affected by seborrhea-like dermatitis with psoriasiform elements [17]. Moreover, in human squamous cell carcinomas (SCCs) (head and neck, esophagus, cervix, and lung), missense and truncating mutations as well as genomic deletions of the ZNF750 locus have been described [18,19]. In addition, ZNF750 expression is low or undetectable in SCC tissues [20]. Remarkably, these genetic and expression alterations were almost exclusively observed in squamous tumors, highlighting the lineage-specific role of ZNF750 in squamous cancer biology. A low expression level of ZNF750 correlates with a higher incidence of undifferentiated histology and is associated with malignant progression and poor prognosis in SCC patients. We have recently reported that ZNF750 inhibits the migratory and invasive properties of breast cancer cells by recruiting the epigenetic platform KDM1A/HDAC1 to the genetic loci of LAMB3 and CTNNAL1, ultimately repressing their expression [21,22].

Although it has already been recognized that prostate cancer progression is dependent on the ability of p63 to control EMT, a process that occurs in different types of cancer and is regulated by multiple mechanisms (i.e., other members of the p53 family or redox regulators), in the last decade several genomic studies focused on the genomic landscape of primary prostate cancer in order to identify other alterations potentially involved in tumor progression [23,24,25,26,27,28,29,30,31,32,33,34,35,36,37,38,39,40,41,42,43,44,45,46,47,48,49,50,51,52,53,54,55,56,57,58,59,60,61,62,63,64,65,66,67,68]. Large chromosomal rearrangements affecting either the most common tumor suppressor gene, including p53 and PTEN, or oncogene, including c-Myc, have been described [23,51]. In addition, several distinct genomic alterations such as ETV1, ETV4, SPOP, FOXA1, and IDH1 are also present in prostate cancer [52]. More recently, a data-driven deep learning approach found that the genes ADIRF, SLC2A5, C3orf86, and HSPA1B are among the most significant prostate cancer biomarkers [53].

However, at the state of art, only a few biomarkers are used to stratify patients according to the risk of developing metastatic lesions or to predict the most appropriate therapeutic strategy in order to optimize the biological response [54].

Starting from all these considerations, we have hypothesized a possible role for ZNF750 in prostate cancer. Specifically, the aim of this study was to investigate whether ZNF750 could have a role in better stratifying patients, identifying those with a higher risk of metastasis and with the poorest prognosis.

## 2. Results

### 2.1. Clinical Features of the Patient Cohort

The mean age of the enrolled patients was 69.8 ± 4.1 for patients with hyperplasia and 72.6 ± 4.8 for patients with acinar prostatic adenocarcinomas. No significant differences were observed.

Metastases were reported in 17 cases (39.5%). In 14 cases, both lymph node and bone metastases were present, while in the remaining 3 cases only bone metastases were observed. As shown in Table 1, metastatic lesions were observed in 5.9% of biopsies classified as grade group 1, 11.8% in grade group 2, 17.6% in grade group 3, and 52.9% and 11.8% in biopsies classified as grade group 4 and 5, respectively.

Table 1 also reports the numbers and percentages of metastatic cases according to the Gleason score.

### 2.2. ZNF750 Expression Is Reduced in Prostate Cancer

The expression of the ZNF750 protein was evaluated by immunohistochemical analysis, and the results are reported in Figure 2 and Table 2.

All patients with benign prostatic hyperplasia (control group) showed marked positivity in all acinar cells. In particular, the intensity of the staining was higher in the nucleus than the cytoplasm. Indeed, ZNF750 positivity was observed in almost all (>80%) nuclei, with a 3+ score, whereas more variability in terms of signal was recorded at the cytoplasmic level, ranging from 40% to 80% of cells, with a 2+/3+ score. Conversely, in prostate acinar carcinomas a significant decrease in ZNF750 positivity in both cytoplasmic and nuclear compartments was observed. Only 6 out of 34 cases showed cytoplasmic positivity (17.6%), whereas 11 samples were completely negative (34.4%) at the nuclear staining. Interestingly, no significant correlation among Gleason score, grade group, and positive or negative ZNF750 cytoplasmic and nuclear expression was noted. However, a positive correlation between the absence of nuclear staining and the presence of metastases was observed (*p* = 0.01). To further confirm the impairment of ZNF750 in prostate cancer, we assessed the expression of ZNF750 both in a normal prostate epithelial cell line (RWPE1) and in prostate cancer cell lines (PC3 and DU145). As shown in Figure 3A,B, both the mRNA and protein levels of ZNF750 are significantly reduced in prostate cancer cell lines compared to the normal cell line. In addition, as expected, ZNF750 is mainly localized in the nucleus in the normal prostate cell line, in accordance with its function as a transcription factor (Figure 3C).

### 2.3. Loss of ZNF750 Nuclear Expression Predicts Risk of Metastatic Prostate Cancer

Logistic regression analysis was applied to identify the risk of prostate cancer metastasis formation associated with the Gleason score, grade group, and absence of ZNF750 nuclear staining. It should also be highlighted that the absence of ZNF750 expression is a risk factor for metastases, regardless of both Gleason score and grade group. Indeed, as reported in Table 3, the odds ratio for the metastases was about 13-fold compared to both Gleason score and grade group; these risks are significantly higher than the traditional predictive risk factors.

### 2.4. Low Levels of ZNF750 Are Associated with a Worse Prognosis

The results obtained from our patient cohort suggest that ZNF750 expression is reduced in patients affected by prostate cancer compared to the control group. To further support these results, a bioinformatics analysis using publicly available datasets of prostate cancer patients was performed. According to our results, the bioinformatics analysis showed that ZNF750 expression is significantly lower (*p* = 8.25 × 10^−12^) in both primary and metastatic tumors compared to normal tissues (Figure 4A,B). Interestingly, the bioinformatics analysis indicates that the promoter region of the ZNF750 gene is significantly hypermethylated (*p* = 4.58 × 10^−9^) in patients affected by prostate cancer, suggesting that one possible mechanism responsible for the downregulation of ZNF750 in cancer is the methylation of the promoter (Figure 4C). However, the levels of ZNF750 promoter methylation are comparable between the cell lines tested (Figure 4D), suggesting that alternative molecular mechanisms are responsible for the downregulation of ZNF750 expression in prostate cancer cell lines.

In human SCCs (head and neck, esophagus, cervix, and lung), missense and truncating mutations as well as genomic deletions of the ZNF750 locus have been described [18,20]. Therefore, we asked whether in prostate cancer ZNF750 results also mutated. To assess the correlation between the expression of ZNF750 and the disease-free survival rate, the GEPIA Dataset was queried. As shown in Figure 4E, low levels of ZNF750 expression are associated with worse disease-free survival. By querying the cBioPortal website, several mutations throughout ZNF750 were identified [55]. All the mutations described are missense mutations (Figure 4F). However, the frequency of mutation is very low (<1%).

## 3. Discussion

In this study, for the first time, a significant reduction in ZNF750 protein expression in acinar prostate carcinomas, both at the cytoplasmic and nuclear level, has been demonstrated. More interestingly, the absence of ZNF750 nuclear staining may represent an important prognostic factor, since it is associated with a markedly increased risk of metastasis regardless of the traditional prostate cancer prognostic factors, such as the Gleason score and grade group [56,57]. In particular, our data demonstrated that the loss of ZNF750 nuclear positivity indicated a 13-fold increase in the risk of prostate metastasis formation with respect to both the Gleason score and grade group. Overall, the results reported here suggest that ZNF750 expression could potentially be a novel and reliable prognostic biomarker, which could be used to recognize prostate cancer with metastatic capacity. Specifically, ZNF750 expression decreases with the increase in tumor aggressiveness, with a full impairment in metastatic lesions. Moreover, our bioinformatics analysis points out that ZNF750 is also mutated in prostate cancer (Figure 4F). All the mutations described are missense mutations. However, those mutations are present in a region of the protein that does not contain biologically active domains. Therefore, whether those mutations affect protein stability or other aspects of protein function remains to be explored [58,59,60]. Another question that still needs to be addressed is which molecular mechanism underlies the loss of ZNF750 expression in cancer. Promoter methylation is one of the main epigenetic mechanisms that plays a significant role in gene silencing [61]. Although the bioinformatics analysis indicates a significant increase in ZNF750 promoter methylation, it should be noted that, according to the literature, the promoter region in normal samples is most likely methylated [62,63]. This consideration is in agreement with the methylation levels observed in our normal prostate epithelial cell lines (RWPE1). In conclusion, at this stage it is very difficult to draw clear conclusions regarding the molecular mechanisms involved in the regulation of ZNF750 expression in cancer. On the other hand, we cannot exclude that DNA methylation does not contribute to the regulation of ZNF750 expression. Indeed, there is experimental evidence indicating that the alteration in DNA methylation in some cancers occurs not at the CpG island within the promoter but at CpG island shores (2 kb distant) [64]. Moreover, additional regulatory mechanisms should also be considered, including promoter sequence variants and m6A-mediated repression of ZNF750 expression [65].

The possible role of ZNF750 as a reliable prognostic biomarker for metastatic prostate cancer is also supported by the literature [66,67]. Indeed, ZNF750 is underexpressed in squamous cell carcinoma, and low levels of ZNF750 are associated with poor survival. In addition, Otsuka et al. revealed that ZNF750 predicts the sensitivity and the response to chemoradiotherapy in esophageal and oral squamous cell carcinoma [66]. Breast cancer shares several biological features such as the hormone dependence for tumor growth, and breast cancer subtypes Luminal A and Luminal B are known to have remarkable biological similarities with prostate cancer [68,69,70,71]. We have previously shown that low expression of ZNF750 predicts a worse disease-free survival compared to patients with high expression, independently from the breast cancer histotype [21]. However, it is reasonable to think that assessing only the expression of ZNF750 in prostate cancer samples is not sufficient for predicting patient prognosis. Therefore, it is possible to speculate that ZNF750 may represent a reliable prognostic biomarker in blood, urine, and formalin-fixed paraffin-embedded prostate tissue, as proposed in recent years [72,73,74].

## 4. Materials and Methods

### 4.1. Case Selection

In this study, 57 consecutive patients, 34 with prostatic acinar carcinoma and 23 with benign prostatic hyperplasia, submitted to prostate mapping biopsy at Rome “Tor Vergata” University Polyclinic were enrolled. The prostate mapping biopsy was performed with at least 16 sample areas of both prostatic lobes. Each fragment was formalin-fixed for 24 h and paraffin-embedded [75]. Serial sections were obtained for morphological and immunohistochemical analysis. For each sample, hematoxylin and eosin staining was performed. The Gleason score and the grade group were assessed according to the 2016 WHO [76]. In addition, according to National Comprehensive Cancer Network clinical practice guidelines (NCCN-g), three general risk groups, based on the prostate-specific antigen (PSA), digital rectal examination (DRE), and biopsy, are recognized to better stratify patients as follows: Low risk: tumor is confined to the prostate, and the PSA is <10 and grade group 1 (Gleason 6). Intermediate risk: tumor is confined to the prostate, and the PSA is between 10 and 20, or grade group 2 or 3 (Gleason 7). This category is often divided into a favorable and unfavorable intermediate risk. High risk: tumor extends outside the prostate, with PSA > 20, or grade group 4 or 5 (Gleason 8 to 10). Lastly, very aggressive tumors are defined as very high risk, in which the tumor has extended into the seminal vesicles (T3b) or the rectum or bladder (T4), or there are multiple biopsy samples with high-grade cancer [77]. Moreover, the tumor size and the presence of lymph node and bone metastases were evaluated. The immunohistochemical study was carried out on serial sections obtained from the most representative primary and secondary Gleason score pattern samples. This study was approved by the Institutional Ethical Committee of the “Policlinico Tor Vergata” (reference number #129.18). Experimental procedures were performed in accordance with The Code of Ethics of the World Medical Association (Declaration of Helsinki). Informed consent was obtained from all subjects involved in the study.

### 4.2. Cell Culture

All cell lines used were obtained from American Type Culture Collection and maintained at 37 °C in 5% CO_2_ in culture medium. Cells were grown using the following medium: PC3, F-12K medium supplemented with 10% FCS (Invitrogen, Waltham, MA, USA) and penicillin/streptomycin 1 U/mL (Gibco, Waltham, MA, USA); DU145, EMEM supplemented with 10% FCS (Invitrogen) and penicillin/streptomycin 1 U/mL (Gibco); and RWPE-1, K-SFM supplemented with bovine pituitary extract (BPE), human recombinant epidermal growth factor (EGF), and penicillin/streptomycin 1 U/mL (Gibco).

### 4.3. Immunohistochemical Analysis

For each case, the expression of anti-ZNF750 was evaluated in both the tumoral primary and secondary Gleason grade area of patients with and without metastasis and in hyperplastic prostatic tissues. Briefly, sections were deparaffinized and, after antigen retrieval (HIER solution at pH 6; 98 °C for 30 min), were incubated with anti-ZNF750 (dilution 1:40, rabbit, polyclonal, Sigma HPA023012) for 1h at room temperature. The UltraTek HRP Anti-Polyvalent Staining System (Scytek, 205 South 600 West Logan, WV, USA) was used for detection. In all cases, the percentage of positive cells was evaluated. Specifically, cytoplasmatic and/or nuclear positivity was defined by a 0–3+ scoring, according to the following criteria: score 0, none or exceptional positive cells; score 1+, ≤10% positive cells; score 2+, 11–50% positive cells; score 3+, >50% positive cells. Histopathologic examination was independently performed by two different pathologists blinded to the clinical data. Interobserver reliability was >98%.

### 4.4. RNA Isolation and Quantitative Real-Time PCR

Total RNA from cells was isolated using RNeasy minikit (Qiagen, Hilden, Germany) according to the manufacturer’s instructions. RNA samples were treated with RNase-free DNase I (Qiagen), and RNA was quantified using a NanoDrop spectrophotometer (Thermo Scientific, Waltham, MA, USA). Total RNA was reverse-transcribed using Superscript III reverse transcriptase and oligo(dT) primer (Invitrogen) [78,79]. qRT-PCR was performed in ABI PRISM 7000 Sequence Detection System (Applied Biosystem, Waltham, MA, USA) with SYBR green ready mix (Applied Biosystem) and specific primers:

ZNF750 for: 5′-AGCTCGCCTGAGTGTGAC-3′;

ZNF750 rev: 5′-TGCAGACTCTGGCCTGTA-3′;

TBP fwd: 5′-TCAAACCCAGAATTGTTCTCCTTAT-3′;

TBP rev: 5′-CCTGAATCCCTTTAGAATAGGGTAG-3′.

Relative mRNA levels were calculated with the 2^−ΔΔCt^ method after normalization to TATA-binding protein (TBP).

### 4.5. Confocal Microscopy

Cells were seeded into 12-well plates onto glass coverslips. After 24 h, cells were fixed in 4% paraformaldehyde and then permeabilized with 0.1% Triton X-100. For the blocking step, 3% BSA/PBS solution was used for 1 h. Then, cells were incubated with ZNF750 antibody (HPA023012) for 4 h and diluted 1:250 in 3% BSA/PBS solution. The incubation with the secondary antibody Alexa Fluor^®^ 488 (ab150077) was performed for 1 h and diluted 1:500 in 3% BSA/PBS solution. Nuclei were stained with DAPI for 15 min. The coverslips were then mounted with ProLong™ Gold Antifade Mountant (Thermo Fisher) onto glass slides and observed under a confocal microscope (Leica Stellaris, Wetzlar, Germany) using the 63× objective.

### 4.6. Promoter Methylation

The promoter methylation analysis was performed using Active Motif’s Bisulfite Conversion Kit, following the manufacturer’s instructions. Briefly, genomic DNA was extracted from cell lines using the Wizard^®^ Genomic DNA Purification Kit and then quantified with NanoDrop™. Samples were then treated with Proteinase K for 30 min at 50 °C. For the conversion reaction, 2 μg of DNA was used. The quality of the converted product was checked using the PCR Primer Mix provided by the kit. PCR was performed using the REDTaq^®^ ReadyMix™ PCR Reaction Mix. Specific primers for methylated and unmethylated DNA are listed below:

ZNF750 Met F1—TTT ATT TTA ATT GCG GAT ATA TCG A;

ZNF750 Met R1—TTA CAC CCA CCG AAC TAC TAC G;

ZNF750 Unmet F1—TTT ATT TTA ATT GTG GAT ATA TTG A;

ZNF750 Unmet R1—TTT ACA CCC ACC AAA CTA CTA CAA A;

ZNF750 Met F2—TTT ATT TTA ATT GCG GAT ATA TCG A;

ZNF750 Met R2—AC ACC CAC CGA ACT ACT ACG A;

ZNF750 Unmet F2—TTT TTA TTT TAA TTG TGG ATA TAT TGA;

ZNF750 Unmet R2—TTT ACA CCC ACC AAA CTA CTA CAA A.

### 4.7. Bioinformatics Analysis and Mutational Analysis

To investigate the potential prognostic value of ZNF750, a disease-free survival curve and differential gene expression analysis were performed. Specifically, the GEPIA Dataset and TNMplot tool of the Kaplan–Meier Plotter were employed to compare the expression of the gene in normal prostate, prostatic cancer, and metastasis. GEPIA Dataset included 492 tumoral and 52 normal samples from TCGA data, whereas TNMplot graph refers to 106 normal samples, 283 prostatic cancer tissues, and 6 metastases. The bioinformatics analysis was carried out by using the online available tools: GEPIA, cBioPortal, Kaplan–Meier Plotter, and UALCAN [80,81,82,83,84,85,86,87]. Mutational analysis was carried out using the cBioPortal website [55].

### 4.8. Statistical Analysis

Data were analyzed using SPSS version 16.0 (SPSS Inc., Chicago, IL, USA) software. Continuous variables were expressed as the mean ± SEM. Categorical data were analyzed using the chi-square test or the Fisher exact test.

The odds ratio for metastasis risks was evaluated for grade group, Gleason score, and ZNF750 expression by logistic regression, using the value of EXP (B), where B represents the logistic coefficient. A 2-tailed *p* value < 0.05 was considered statistically significant.

For bioinformatics analysis, the following tests were used: gene expression: ANOVA test, Kruskal–Wallis test, and Dunn’s test; promoter methylation: Welch’s *t*-test; survival analysis: log-rank test.

## 5. Conclusions

In conclusion, it is emerging that ZNF750 can potentially act as a prognostic biomarker in cancer and most likely be used to predict the formation of metastasis. Risk assessment is of fundamental importance for clinical decision making. Therefore, a better stratification of patients by both prognostic and predictive factors may lead to a higher-quality management of the disease and to the identification of new therapeutic strategies.

## Figures and Tables

**Figure 1 ijms-24-06519-f001:**
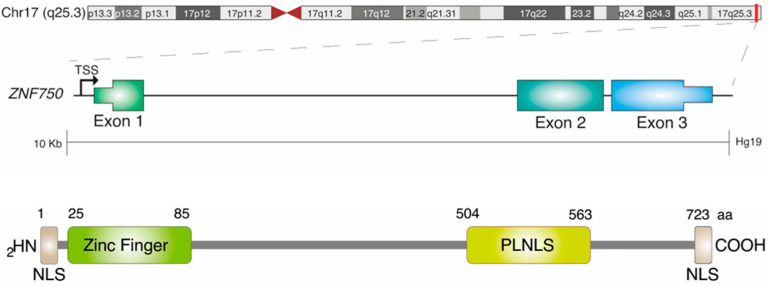
Schematic representation of ZNF750 gene organization and protein structure. ZNF750 gene is located on chromosome 17. The structural organization of ZNF750 consists of an atypical C2H2 zinc finger motif in the amino terminal domain, which is required for the binding of ZNF750 to DNA and regulating gene transcription. In addition, two highly conserved PLNLS sequences that are required for protein–protein interaction are present in the carboxy terminal domain.

**Figure 2 ijms-24-06519-f002:**
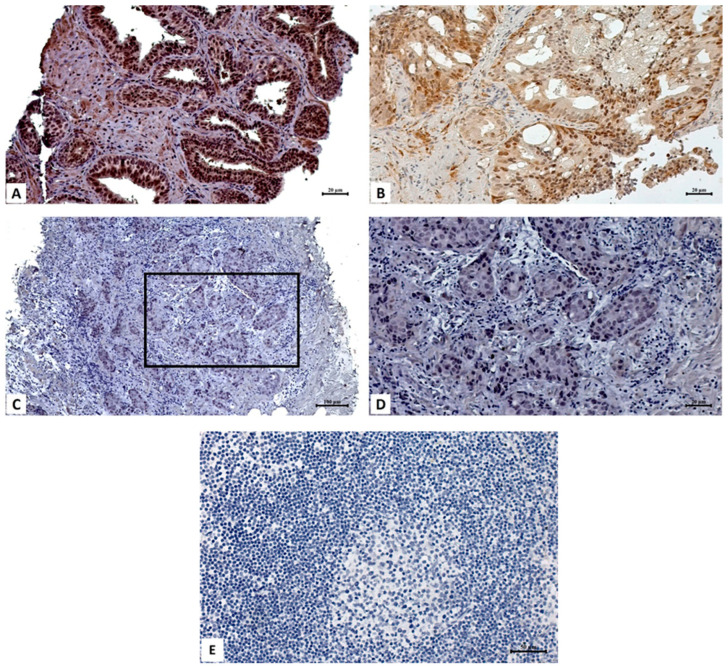
Immunohistochemical analysis. (**A**) Image shows high ZNF750 protein expression in both nucleus and cytoplasm of prostatic hyperplasia cells (scale bar represents 20 µm). (**B**) Non-metastatic prostatic acinar adenocarcinoma, Gleason 8 (4 + 4), characterized by high nuclear and weak cytoplasmic ZNF750 expression (scale bar represents 20 µm). (**C**) No ZNF750 expression in metastatic prostatic acinar adenocarcinoma, Gleason 8 (4 + 4) (scale bar represents 100 µm). Square represents the high magnification in panel (**D**). (**D**) Absence of ZNF750 staining in both nuclear and cytoplasmic compartments (scale bar represents 20 µm). (**E**) ZNF750 staining performed on lymph node tissue as a negative control for ZNF750 immunohistochemical reaction (scale bar represents 50 µm).

**Figure 3 ijms-24-06519-f003:**
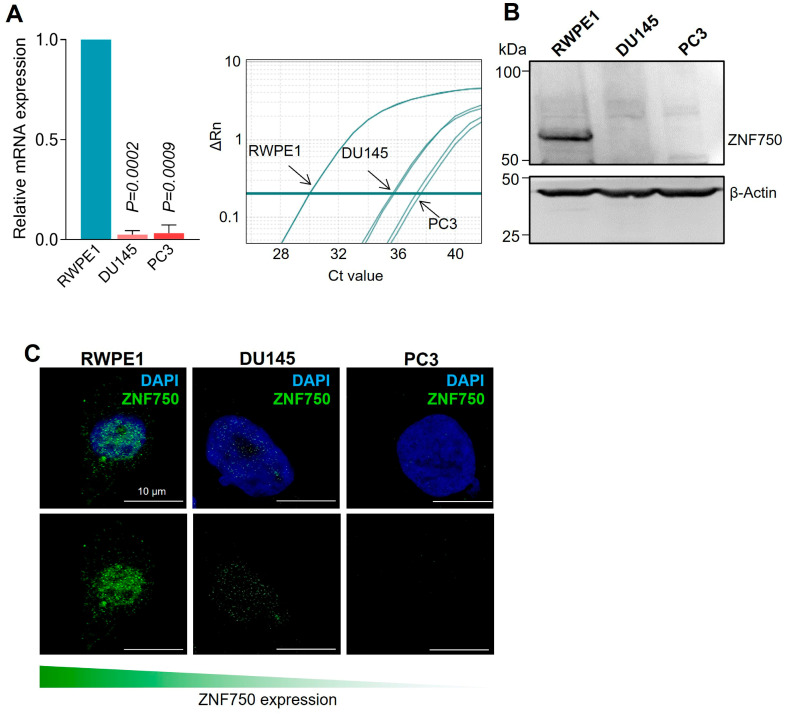
ZNF750 expression is reduced in prostate cancer cell lines. (**A**) ZNF750 mRNA expression in the indicated cell lines assessed by real-time PCR assay. (**B**) ZNF750 protein levels were evaluated by Western blot. A representative image of three independent experiments is shown. (**C**) Confocal microscopy analysis showing that ZNF750 is mainly located in the nucleus of normal prostate epithelial cells. Bars represent means ± SD of three independent experiments.

**Figure 4 ijms-24-06519-f004:**
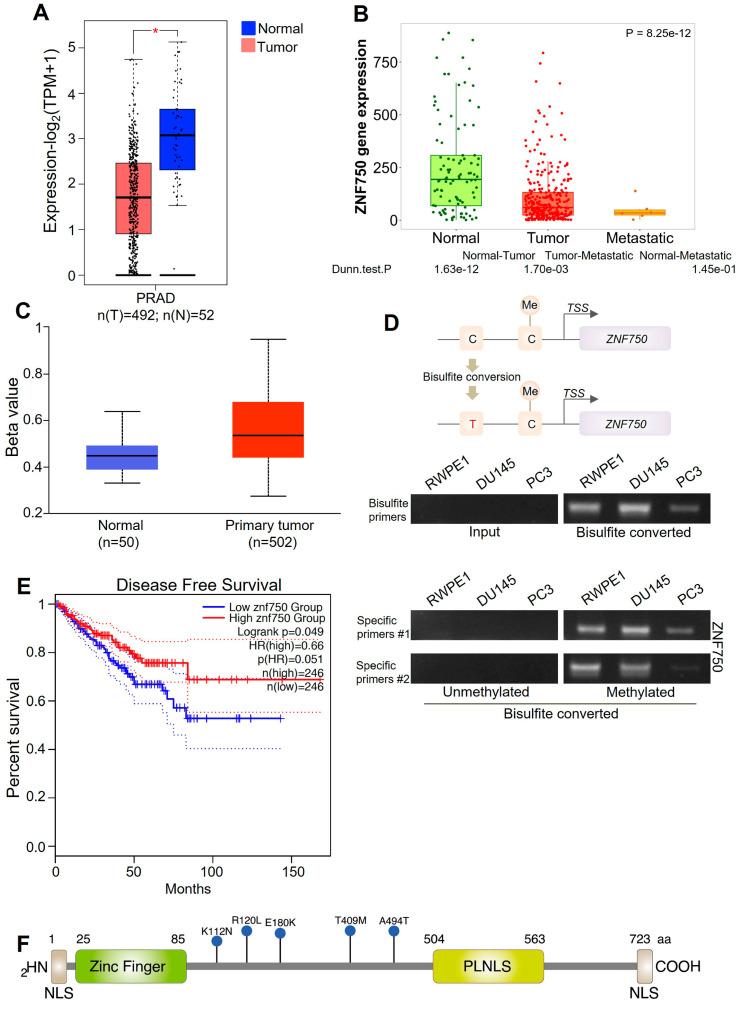
Bioinformatics analysis. (**A**) ZNF750 mRNA expression in tumoral (red, n = 492) and normal (blue, n = 52) tissues, http://gepia.cancer-pku.cn/index.html (accessed on 15 December 2022). * *p*-value < 0.05 (**B**) Differential expression of ZNF750 mRNA in normal (n = 106), tumor (n = 283), and metastatic tissues (n = 6) in prostate cancer, https://tnmplot.com/analysis/ (accessed on 15 December 2022). (**C**) ZNF750 promoter region is methylated in prostatic acinar adenocarcinomas (normal, n = 50 and primary tumor, n = 502). (**D**) ZNF750 promoter methylation status both in normal and cancer cell lines. (**E**) Patients with low expression of ZNF750 mRNA showed a shorter disease-free survival than patients with high expression of ZNF750 (high n = 246 and low n = 246), http://gepia.cancer-pku.cn/index.html (accessed on 15 December 2022). See Section 4 for details of statistical analysis. (**F**) The most common mutations described in human prostate cancer specimen. Blue lollipops indicate the somatic mutations described in human prostate cancer. The analysis was performed using cBioPortal for Cancer Genomics (http://www.cbioportal.org/ (accessed on 15 December 2022)). NLS, nuclear localization.

**Table 1 ijms-24-06519-t001:** Number of acinar prostatic adenocarcinomas divided at diagnosis by grade group and Gleason score.

Grade Group/Gleason Score	Cases No. (%)	Metastasis No. (%)
1/6 (3 + 3)	4 (9.3%)	1 (5.9%)
2/7 (3 + 4)	13 (30.2%)	2 (11.8%)
3/7 (4 + 3)	7 (16.3%)	3 (17.6%)
4/8 (4 + 4)	17 (39.5%)	9 (52.9%)
5/9 (4 + 5)	2 (4.7%)	2 (11.8%)
Tot.	43 (100%)	17 (100%)

**Table 2 ijms-24-06519-t002:** (A) Evaluation of ZNF750 cytoplasmatic staining in normal prostate and prostatic acinar adenocarcinomas of patients with and without metastasis. (B) Evaluation of ZNF750 nuclear staining in normal prostate and prostatic acinar adenocarcinomas of patients with and without metastasis.

**(A)**			
	**ZNF750 Cytoplasmatic Staining**		
	**Negative**	**Positive**	** *p* **
Benign Hyperplasia	0 (0%)	23 (100%)	0.001
Acinar Carcinoma	28 (82.4%)	6 (17.6%)	
Gleason Score			0.33
6	2 (5.8%)	2 (22.2%)	
7	17 (50.0%)	3 (33.3%)	
8	13 (35.4%)	4 (44.5%)	
9	2 (8.8%)	0 (0%)	
Grade Group			0.37
1	2 (5.9%)	2 (22.2%)	
2	12 (35.3%)	1 (11.1%)	
3	5 (14.7%)	2 (22.2%)	
4/5	15 (44.1%)	4 (44.5%)	
Metastasis			0.006
with	17 (100%)	0 (0%)	
*w*/*o*	17 (65.4%)	9 (34.6%)	
**(B)**			
	**ZNF750 Nuclear Staining**		
	**Negative**	**Positive**	** *p* **
Benign Hyperplasia	0 (0%)	23 (100%)	0.001
Acinar Carcinoma	11 (34.4%)	21 (65.6%)	
Gleason Score			0.16
6	0 (0%)	4 (12.9%)	
7	4 (33.3%)	16 (51.6%)	
8	6 (50.0%)	10 (32.3%)	
9	2 (16.7%)	1 (3.2%)	
Grade Group			0.09
1	0 (0%)	4 (12.9%)	
2	3 (25.0%)	10 (32.3%)	
3	1 (8.3%)	6 (19.3%)	
4/5	8 (66.7%)	11 (35.5%)	
Metastasis			0.01
with	10 (58.8%)	7 (41.2%)	
*w*/*o*	2 (7.7%)	24 (92.3%)	

**Table 3 ijms-24-06519-t003:** Odds ratio of risk of metastasis in patients with prostatic acinar adenocarcinomas.

	*p*	Odds Ratio (EXPB)	95% C.I.
Grade Group	0.10	1.89	0.88–4.10
Absence of ZNF750 nuclear positivity	0.005	13.40	2.22–80.81
Gleason Score	0.33	1.70	0.58–4.98
Absence of ZNF750 nuclear positivity	0.004	13.71	2.32–81.02

## Data Availability

Bioinformatics data can be found at: http://gepia.cancer-pku.cn/index.html; https://www.cbioportal.org; https://kmplot.com/analysis/; accessed on 15 December 2022. Histological, immunohistochemical, and molecular data will be provided on request.

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
