# Peer review of "ZNF750: A Novel Prognostic Biomarker in Metastatic Prostate Cancer"

_ijms, 2023, doi:10.3390/ijms24076519_

Round 1
Reviewer 1 Report
In this manuscript, Montanaro et al. report that ZNF750 might represent a novel prognostic biomarker in metastatic prostate cancer. The presented work is potentially important for a better stratification of patients with PC, relevant to the scope of International Journal of Molecular Sciences, and will be of interest to its readership. However, there are some issues that need to be addressed as detailed below.
Figure 2: Protein expression data should be supported by mRNA expression data.
Figure 4: Table labeling should be corrected.
In Figure 4D data should be supported by qRT-PCR analysis or at least a relative quantification should be shown.
Author Response
Dear Editor,
Thank you for the opportunity to resubmit our manuscript entitled "ZNF750: A Novel Prognostic Biomarker in Metastatic Prostate Cancer" (ijms-2238863) to the International Journal of Molecular Sciences. We appreciate the time and effort of the reviewers in providing us further interesting suggestions and insights.
In response to the previous revision (ijms-1817827), we made significant corrections, incorporated new experimental data, and addressed the reviewers' recommendations, which have greatly improved the quality of the study. According to this, we believe that the current version of the manuscript adequately addresses the concerns raised in the first revision round.
However, we received a second round of "major revision" that questions our evidence and the conclusions made by the previous reviewers. The latest referee report suggests that our findings are insufficient to support the conclusions, and additional time-consuming and expensive experiments using mouse metastatic models or Patient-Derived Xenografts have been requested.
Moreover, we would like to point out that such experimental procedures cannot be conducted within ten days.
While we acknowledge that in-vivo experiments would contribute to validating this molecule as a reliable prognostic marker for clinical use, it is beyond the scope of our paper, which aims to explore the potential prognostic value of ZNF750 in identifying patients at higher risk of metastasis.
According to these considerations, we respectfully request that the Editor reconsider our study for publication in the International Journal of Molecular Sciences in its current version. If this request cannot be satisfied, we will withdraw the manuscript for submission to another journal.
Reviewer 2 Report
The objective of this manuscript is to identify ZNF750 as a new prognostic biomarker in metastatic prostate cancer. The authors find that there is a correlation between ZNF750 nuclear staining and metastases. If the author could approve the point, this study would be very valuable and helpful for the prognosis in metastatic prostate cancer. However, in my opinion, the authors don't show enough evidence to support their conclusions:
1.To demonstrate whether ZNF750 could be regarded as a prognostic biomarker, the author should set up some experiments in mouse metastatic model to compare ZNF750 expression (especially in nuclear) and function before/after metastasis.
2.The evidence in Fig.2 and 3 could not sufficiently support the conclusions. So, could the authors further identify ZNF750 expression pattern in prostate cancer with or without metastasis by using PDX model(PMCID: PMC5354949)?
3. Could the authors provide some evidence to test the relationship between ZNF750 and EMT markers?
4. The data from manuscript are not enough to support ZNF750 as a prognostic biomarker. The author should provide more evidence to support the conclusion.
5.In Fig.2, the authors should also identify the cancer cells by cancer or epithelial markers. Then, it would be more reasonable by double staining in immunohistochemical analysis.
6.The quality of western blot in Fig.3B is not good.
Author Response

(The authors gave the same response as above.)

Round 2
Reviewer 2 Report
It is valuable for the prostate cancer prognosis to identify ZNF750 as a new prognostic biomarker in metastatic prostate cancer. I can accept the current version. Some minor errors should be addressed. For example: the quality of western blot in Fig.3B is not good. The figure should be replaced. In Fig.4, the labels should be clearer.
Author Response
Ref: ijms-2238863.
"ZNF750: a novel prognostic biomarker in metastatic prostate cancer"
Submitted to: International JOurnal of Molecular Science
Before we begin the point-by-point review of the list of concerns, we would like to thank the Reviewer for comments on how to improve the manuscript, which has been revised accordingly, as well as the Editors for calling for a new submission of an improved version of our manuscript.
Reviewer#1
It is valuable for the prostate cancer prognosis to identify ZNF750 as a new prognostic biomarker in metastatic prostate cancer. I can accept the current version. Some minor errors should be addressed. For example: the quality of western blot in Fig.3B is not good. The figure should be replaced. In Fig.4, the labels should be clearer.
Reply: we would like to thank the Reviewer for expressing interest in our work, and for their availability to review our manuscript. In the new version of our manuscript we modified the figures 3 and 4 according to the reviewer' comments.